# Low Muscle Mass Is Associated with Poorer Glycemic Control and Higher Oxidative Stress in Older Patients with Type 2 Diabetes

**DOI:** 10.3390/nu15143167

**Published:** 2023-07-17

**Authors:** Blanca Alabadi, Miguel Civera, Adrián De la Rosa, Sergio Martinez-Hervas, Mari Carmen Gomez-Cabrera, José T. Real

**Affiliations:** 1Service of Endocrinology and Nutrition, Hospital Clínico Universitario of Valencia, 46010 Valencia, Spain; mi.civeraa@comv.es (M.C.); sergio.martinez@uv.es (S.M.-H.); jtreal@uv.es (J.T.R.); 2INCLIVA Biomedical Research Institute, 46010 Valencia, Spain; 3CIBER de Diabetes y Enfermedades Metabólicas Asociadas (CIBERDEM), ISCIII, 28029 Madrid, Spain; 4Department of Medicine, University of Valencia, 46010 Valencia, Spain; 5Laboratory of Applied Sciences of Sport and Innovation Research Group (GICED), Unidades Tecnológicas de Santander (UTS), Bucaramanga 680006, Colombia; adrian1031@gmail.com; 6Freshage Research Group, Department of Physiology, Faculty of Medicine, University of Valencia, 46010 Valencia, Spain; carmen.gomez@uv.es; 7CIBER de Fragilidad y Envejecimiento Saludable (CIBERFES), ISCIII, 28029 Madrid, Spain

**Keywords:** muscle mass, diabetes, aging, body composition, oxidative stress

## Abstract

Body composition changes that occur during aging, such as loss of lean mass, are unfavorable at metabolic level and they can explain, in part, the appearance of certain age-associated diseases such as type 2 diabetes (T2D). Separately, T2D is associated with an increase in oxidative stress (OS) which negatively affects skeletal muscle. Our aim was to study the differences in clinical and nutritional parameters, disease control, and OS in a cohort of older patients with T2D classified according to the amount of lean mass they had. We included 100 adults older than 65 years with T2D. We found that women with low fat-free mass and muscle mass have worse T2D metabolic control. Moreover, the patients with a low percentile of muscle mass present a high value of OS. The study shows that the presence of low lean mass (LM) in the geriatric population diagnosed with T2D is associated with poorer glycemic control and greater OS.

## 1. Introduction

Aging is a physiological process that leads to a functional decline of cells, tissues, and organs. It is a complex phenomenon that comprises different changes during the progression of life, such as increased inflammation, oxidative stress (OS), DNA mutations, mitochondrial dysfunction, accumulation of senescent cells, and loss of proteostasis [1].

The study of the aging process has become important due to the increase in life expectancy and because it is an ideal scenario for the development of age-related diseases, such as cancer, cardiovascular diseases, or metabolic dysfunctions observed in the metabolic syndrome, a group of disorders that includes type 2 diabetes (T2D) [2].

Limitations in basic or instrumental activities of daily living, also known as functional impairment, is a characteristic phenotypic expression of diseases in older adults and has great clinical relevance, since as a person ages, the prognostic value of the diseases decreases and that of functional status increases [3]. Functional deterioration is strongly related to adverse factors such as institutionalization [4] or mortality [5], and certain factors are known to increase risk, such as female gender, age, or the presence of chronic diseases such as type 2 diabetes [6]. In addition, it is linked to changes in body composition.

Loss of skeletal muscle mass (SMM) is one of the first symptoms related to tissue aging and has an important impact on metabolism and body homeostasis. The loss of SMM and function is between 5–13% in the population of 60–70-year-olds and increases to 11–50% in the case of those over 80 years of age as a result of progressive atrophy, caused by loss of muscle fibers, reduced motor neuron input, and impaired function of contractility within each fiber [7].

In addition to the loss of lean mass (LM), the increase in body fat, especially at the abdominal level, and the loss of bone tissue [8] are other examples of the most important changes in body composition that appear progressively during aging and, according to multiple studies, these modifications are related to a worse quality of life [9]. These changes are unfavorable at the metabolic level and can explain, in part, the appearance of certain age-associated diseases such as T2D [10]. Inflammation, OS, mitochondrial dysfunction, the presence of malnutrition, and different energy imbalances are also implicated in their development [11,12].

The relationship between body composition and T2D is bidirectional. As described, changes in body composition associated with aging increase the risk of developing T2D. In turn, T2D is associated with an increase in OS, and this negatively affects skeletal muscle, an organ with a fundamental role in this disease since it is involved in the regulation of glucose metabolism [13], as shown in Figure 1.

As has already been shown, diabetic patients present worse functionality as well as a higher OS [14]. However, there are few studies that analyze the relationship between the amount of LM and SMM with glycemic control and OS. Therefore, it is of great interest to know how those older diabetic patients who have very low LM behave clinically.

In this sense, since the prevention of disabilities and the preservation of the quality of life of the elderly is one of the main health objectives, the aim of our study was to study the differences in clinical and nutritional parameters, disease control, and OS in a cohort of older adults with T2D classified according to the amount of LM they had. With this, it would be possible to identify that if in a diabetic population, the amount of functional mass affects the pathology control and that it influences the metabolic and functional deterioration, thus opening a therapeutic window.

## 2. Materials and Methods

### 2.1. Participants

The included subjects’ characteristics have been previously described [14]. Briefly, we consecutively selected a total of 100 elderly patients (48 men and 52 women) diagnosed with T2D in the outpatient clinic of our center by opportunistic method. Figure 2 contains data on participant screening and enrollment.

The inclusion criteria were as follows: age ≥ 65 years and diagnosis of T2D according to the ADA criteria [15]. In addition, it was required to present a glycated hemoglobin (HbA1c) < 9%, to be treated with metformin and not to be under treatment with allopurinol at the time of inclusion due to the possible implications of these drugs on body composition and OS.

The exclusion criteria were as follows: severe chronic complications of diabetes, systemic diseases, active oncological disease, severe chronic obstructive pulmonary disease, uncontrolled hypothyroidism, cirrhosis, dementia, alterations or serious disorders of fluid regulation, lower limb amputations, or having received recent treatment with oral corticosteroids for more than 30 days.

The study was approved by the Ethics Committee of the Hospital Clínico Universitario de Valencia and all patients gave their informed consent to participate in the study.

### 2.2. Clinical and Anthropometric Parameters

In the study protocol, were determined the follow clinical parameters: years of disease evolution, habitual pharmacological treatment, and nutritional status through MNA^®^ screening (Mini Nutritional Assessment) [16]. In addition, using a food frequency questionnaire and a 24 h dietary recall, the weekly protein intake was determined.

Anthropometric parameters were determined using standardized procedures: weight (kg), height (m), body mass index (kg/m^2^), waist circumference (midpoint between the last rib and iliac crest, in centimeters), calf circumference (most prominent part of the gastrocnemius muscle, in centimeters), and brachial circumference (midpoint between the acromion and olecranon, in centimeters).

An ergonomic, flexible, and inextensible Cescorf tape was used to measure circumferences, and all measurements were performed by the same researcher.

### 2.3. Body Composition and Functionality Parameters

Body composition was determined in all patients after 12 h of fasting and without having performed physical exercise in the last 8 h. A NUTRILAB^TM^ single-frequency bioelectric impedance (Akern s.r.l., Firenze, Italy) was used following the widely accepted methodology [17,18].

Physical performance and muscle strength were used to assess the functionality of the volunteers. Physical performance was assessed by having a walking speed over a 4 m distance, taking the lowest of 3 measurements. Muscle strength was assessed by grip strength with a Jamar^®^ Plus+ digital dynamometer (Patterson Medical, Warrenville, IL, USA), following the recommendations of the American Society of Hand Therapists (ASHT) [19] and taking the maximum of three measurements separated 1 min from each other.

### 2.4. Biochemical Parameters

After 12 h of fasting, blood samples were taken by puncture of an antecubital vein. The sample was divided into two, one part was preserved in EDTA in a BD Vacutainer tube (Stockholm, Sweden, Ref 367525) and used for the analysis of OS parameters. The other part was used for the determination of biochemical parameters by standardized laboratory methods.

Glycemia was determined by enzymatic method (Robonik Prietest Touch Plus ECO, Biochemistry Analyzer, Ambernath, India) [20] and HbA1c by high-performance liquid chromatography (HPLC) [21].

### 2.5. Oxidative Stress

To extract the plasma from the blood sample, the tube was centrifuged at 2000× *g* RCF for 15 min at 4 °C (JP Selecta S.A.; Barcelona, Spain). The supernatant was stored at −20 °C until it was analyzed.

The OS parameters determined in the serum sample were malondialdehyde (MDA) and protein carbonylation. Plasma lipid peroxidation was determined following a method that is based on the hydrolysis of lipoperoxides in plasma and subsequent formation of an adduct between thiobarbituric acid and MDA (thiobarbituric acid–MDA2) [22]. This adduct was detected using high-performance liquid chromatography in reverse phase and quantified at 532 nm (DIONEX Ultimate 3000 chromatograph; Thermo Scientific, Waltham, MA, USA). The procedure to quantify total protein carbonyls was using the OxyBlotTM Protein Oxidation Detection Kit (Ref. s7150; Millipore Corporation, Burlington, MA, USA) (Ref. s750, Millipore, Darmstadt, Germany) and Ponceau S red stain (PanReac AppliChem, Barcelona, Spain; Lot: 6T1226612266) followed by finding the ratio between the total density in the Oxyblot and the Ponceau.

### 2.6. Statistical Methods

The sample size was calculated using the free software G*Power (version 3.1.9.6; Heinrich-Heine-Universität Düsseldorf, Düsseldorf, Germany). A standard alpha error of 5%, a beta error of 20%, and an effect size of 0.6 were taken into account. The minimum n obtained for the entire group was 72 patients.

Data analysis was performed using the Statistical Package for the Social Sciences (SPSS 26 for iOS, SPSS Chicago, IL, USA). For each of the variables, the values are shown as mean ± standard deviation. The *p*-values were two-sided and values lower than 0.05 were considered significant.

The 25th percentile was calculated by gender for the different body composition parameters and the patients were grouped into two groups according to whether they had a ≤25th percentile or >25th percentile.

Subsequently, the parametric Student’s *t*-test or the non-parametric Mann–Whitney U test were performed according to normality for the comparative statistical analysis between these groups for the different determined variables.

Spearman’s correlation was studied as well as multivariate correlation using linear and logistic regressions in order to continue studying the association between low lean mass and poor glycemic control. Independent variables having a significant correlation with a dependent variable (low lean mass) were candidates for multivariable regression.

## 3. Results

One hundred patients were included: 48 men and 52 women. The clinical, anthropometric, body composition and functionality, biochemical, and OS characteristics of the patients, both in the complete cohort and by gender, are described in Table 1 and Table 2, respectively [14].

This is a long-term diabetic population, with good glycemic control, a good nutritional status, and with average anthropometric parameters that place them in type I obesity. Women have a higher proportion of body fat and a lower proportion of fat-free mass (FFM) and SMM than men, in addition to worse physical performance, less muscle strength, and lower protein intake.

As shown in Table 3, when studying the patients according to the fat-free mass index (FFMI), it is observed how, in the complete cohort, statistically significant differences are obtained in brachial circumference (BC), calf circumference (CC), and protein intake among the participants with a ≤25th percentile and the participants with a >25th percentile of the FFMI. In all cases, the highest values were obtained in patients with a higher FFMI.

In the case of women, those with lower FFMI values have a smaller BC (30.1 ± 3.5 vs. 32.7 ± 3.7 cm), a lower protein intake (18.3 ± 4.8 vs. 21.0 ± 4.1%), and a worse glycemic control assessed by fasting blood glucose (178.3 ± 41.1 vs. 136.8 ± 45.2 mg/dL) and glycosylated hemoglobin (7.8 ± 1.1 vs. 7.2 ± 0.9%) compared to those with a FFMI above the 25th percentile. Men, on the other hand, only present statistically significant differences in anthropometric parameters: diabetics with a low FFMI present lower BC (28.2 ± 1.2 vs. 31.4 ± 2.7 cm) and CC (34.6 ± 1.9 vs. 38.7 ± 2.6 cm) than those with a higher index.

On the other hand, when assessing the patients according to their amount of SMM, both expressed in kg and in index, the results shown in Table 4 and Table 5 are obtained.

In the first case, in the complete cohort statistical significance is obtained in the BC and MDA parameters. Participants with a ≤25th percentile, compared to those with a >25th percentile, present a smaller BC and a greater OS.

When studying men and women separately, it is observed how women with a lower amount of SMM have worse glycemic control assessed by HbA1c (7.9 ± 1.1 vs. 7.2 ± 0.9%) and a higher OS assessed by MDA (9.0 ± 7.1 vs. 5.0 ± 4.7 µM) than those with a higher amount of SMM. As for men, those with a low SMM percentile have a lower BC (29.0 ± 2.7 vs. 31.4 ± 2.9 cm), a lower CC (35.4 ± 2.6 vs. 38.6 ± 2.8 cm), and a higher OS measured by MDA (9.1 ± 7.7 vs. 4.7 ± 4.9 µM) than those with a higher percentile of SMM.

In the second case, when studying the patients according to the skeletal muscle 222 mass index (SMI), those participants with a ≤25th percentile present smaller CC and 223 higher OS valued by the MDA.

When we divided the cohort by gender, it was observed how women with lower SMI have a higher HbA1c (7.8 ± 1.1 vs. 7.2 ± 0.9%) and greater OS (8.4 ± 6.6 vs. 5.1 ± 5.0 µM) than those with a higher SMI. Men present statistically significant differences in BC (29.0 ± 2.5 vs. 31.7 ± 2.8 cm), CC (35.3 ± 2.2 vs. 39.0 ± 2.7 cm), and protein intake (18.0 ± 3.1 vs. 20.6 ± 4.1%), with higher values obtained in men with a >25th percentile in the three parameters.

The univariate analysis shows that the FFMI percentile correlates with BC (r = 0.29) and glycemia (r = −0.41) in women, while in men, it correlates with BC (r = 0.54) and CC (r = 0.58). Regarding muscle mass, the SMI percentile only correlates with MDA (r = −0.37) in the case of men, while the SM percentile correlates with MDA (r = −0.29) and HbA1c (r = −0.30) in women, and BC (r = 0.36) and CC (r = 0.42) in men.

Finally, a multivariate analysis was carried out about women to evaluate the presence of independent relations between low lean mass and poor glycemic control (Table 6). There was not any significant relation between FFMI (Table 6A) and SMM (Table 6B) with the parameters studied.

## 4. Discussion

Population aging has led to an increase in the study of the relationship between functional status, specifically the role of musculoskeletal tissue, with highly prevalent chronic diseases such as T2D. Although the main risk factors for the development of diabetes are still under investigation, muscle mass is the main tissue that contributes to insulin-mediated glucose disposal, so its decrease contributes to insulin resistance [23].

Within the diabetic population, a large 12-year prospective cohort study published in 2017 demonstrated that low muscle mass was strongly associated with an increased risk of T2D independently of obesity in middle-aged adults [24]. Even though there are not many longitudinal studies designed exclusively for the assessment of changes in the body composition of diabetic older adults compared to non-diabetics, in 2009, it was evidenced that in a cohort of 2675 subjects followed up for 6 years, that T2D was associated with a loss of muscle mass [25]. These results have been subsequently repeated in a similar way in other clinical studies, generally being the lean appendicular mass the main affected [10,26,27].

Fat-free mass represents, as its name suggests, all body content except adipose tissue: SMM, bone mass, body organs, and total body water [28]. There is a direct linear correlation between FFM and SMM [29] and that is why it is used as a parameter for assess patient’s functional tissue. It is known that during body weight loss processes, as well as weight maintenance processes, a high protein diet preserves or increases FFM, reduces FM, and improves the metabolic profile [30]. Similar data are obtained for SMM [24].

Sufficient protein intake has classically been placed for healthy adults at around 10–15% of total caloric intake [31]. However, studies such as Protein Summit 2.0 or PROT-AGE argue that an increase in protein intake could benefit elderly health [32], so the recommendation for dietary protein would generally increase to 15–20% of total calories in healthy elderly diet [33].

The diabetic population studied presents a sufficient protein intake and confirms the previously mentioned data, since individuals with a low percentile of FFMI (in the complete cohort and in women) and of SMI (in men) are those with a lower protein intake.

Although there are different studies that link the role of LM with the diagnosis of T2D, the relationship between hyperglycemia per se with low LM, beyond the sarcopenic range, has not been fully investigated [26]. It seems a bidirectional relationship where, in the first place, presenting a low LM is a risk factor for hyperglycemia due to a decrease in glucose clearance. It is known that skeletal muscle is responsible for more than 80% of glucose uptake after an oral glucose load, so desensitization of this organ leads to insulin resistance and elevated blood glucose levels [34].

The prolonged exposure of human cells and tissues to hyperglycemia, on the other hand, promotes the accumulation of advanced glycosylation end-products in skeletal muscle, causing increased OS, mitochondrial dysfunction [35,36], and loss of the anabolic action of insulin [37]. All of this leads to muscle damage and physical inactivity that lead to a loss of muscle mass and function, known as sarcopenia [38].

In our study, despite the fact that it is a cohort with a conserved LM as described in a previously published article [14], the obtained results go along the same lines in women: those with a low FFMI, SMM, and SMI have worse T2D metabolic control, despite the fact that when their relationship is studied in combination in a model, it disappears. Our findings are similar to those obtained by other authors such as Srikanthan et al., who found an inverse association between SMM and insulin resistance [39]; or The Florey Adelaide Male Aging Study, which found an inverse relationship between the syndrome metabolism and SMM in older men [40]. According to a recent meta-analysis, three studies that used body mass index, SMI, or appendicular skeletal muscle mass as measures of muscle mass showed that poor glycemic control was strongly associated with loss of muscle mass but not with loss of muscle function [41]. It is known that each 10% increase in SMI is associated with a 12% reduction in prediabetes or established diabetes [42].

The role of gender, on the other hand, remains unclear in the literature. While in the cohort of this study, a relationship between LM and glycemic control was only obtained in women, there are other studies in which this relationship between T2D and SMM only appears in men [42,43].

Interventions aimed at increasing SMM counteract the development of insulin resistance; however, the mechanism for this is still not fully known [44]. Hyperglycemia, therefore, could be a potentially modifiable factor associated with low LM and future goals could be focused on its control to preserve functional status, although more research is required to clarify this relationship.

In recent decades, numerous groups of professionals dedicated to nutrition and geriatrics have studied a possible intervention aimed to increase muscle mass or improve its functionality in older adults. Leucine supplementation presents the greatest scientific evidence in this population since its significant effect on muscle mass has been demonstrated in different studies [45,46]. A recent review concluded that there is sufficient evidence to recommend leucine supplementation for sarcopenic older people to increase muscle mass, but not for muscle strength or physical performance [47]. Other interventions with evidence in increasing muscle mass are protein supplementation combined with resistance training [48], especially in the obese population, and creatine supplementation combined with resistance training [49].

In a large part of the population, T2D is associated with obesity [50], so in addition to the increase in muscle mass, in many cases older adults can benefit from weight loss that proportionally increases muscle index. In recent clinical research, low-carbohydrate and high-fat diets are one of the most common dietary remedies for patients with obesity and diabetes [51]. A recent review recommends the use of the ketogenic diet, a diet that replaces glucose sugar with ketone bodies, in the treatment of these two pathologies [52].

These interventions also influence OS, although more studies are needed to clarify this role. Flaim et al. demonstrated that the use of a whey protein supplement for 12 weeks in individuals with T2D managed to reduce OS parameters [53]. However, in another study where the MDA and protein carbonylation parameters were analyzed, no changes were found in the supplemented group compared to the control group [54]. Ketosis, although the exact mechanisms are still not well understood, is known to also reduce overall OS [55,56].

In the population studied, the values of the determined OS parameters, MDA, and protein carbonylation, are high compared to values obtained by other authors in the geriatric population [57,58]. These data could be explained by the presence of a body mass index corresponding to type I obesity and T2D as the underlying disease in the cohort, two pathologies that are associated with a higher OS [59].

In addition, sarcopenia shares OS with T2D as a pathophysiological mechanism. The relationship between loss of SMM and function with OS is complex and can be explained by different hypotheses, such as the damage caused by OS at the musculoskeletal system level due to an increase in intracellular calcium [60], the increased production of reactive oxygen species that causes a decrease in myoblasts and apoptosis of muscle cells [61] or the immune activation that OS can cause through oxidized cellular components [62].

In our study, the obtained results confirm the relationship between OS assessed by MDA and SMM. When studying the cohort according to their SMM, both in the complete group and by gender, it is observed how patients with a percentile of this functional parameter than ≤25th present a high value of OS. The same occurs, except in the male group, when studying SMM expressed as an index.

In a previous analysis carried out in the same cohort, it was observed that the OS assessed by protein carbonylation was elevated in frail patients compared to pre-frail and robust patients, with a gradual increase among the three categories of frailty. However, when assessing OS using MDA, different results were obtained, observing an increase in OS in pre-frail patients compared to robust patients above the values of frail diabetics. In addition it was seen how, unlike what happened with SMM, FM increased gradually with frailty, being the only body composition parameter with differences between categories [14].

This relationship between protein carbonylation and frailty, and the lack of relationship between frailty with the amount of SMM, could explain the lack of relationship between OS assessed by protein carbonylation and LM in our study. Cumulative carbonylation of proteins has been shown to affect the structure and function of proteins [63], so this OS parameter could be more closely related to other aspects involved in the diagnosis of frailty and sarcopenia, such as muscle strength and physical performance [64] than with the loss of LM itself in our cohort.

The obtained results are clinically relevant since they highlight the importance of assessing body composition and muscle tissue in a pathology as prevalent as T2D due to the clinical implications it entails. Low SMM itself is associated with decreased muscle strength, increased fatigability, and decreased exercise capacity, along with decreased quality of life [65], while a diagnosis of sarcopenia is known to imply an increase in adverse outcomes such as falls, functional impairment, frailty, and mortality [13]. Additionally, the present study reveals the worse glycemic control and the greater OS presented by patients with a lower functional compartment, with women being mainly affected.

All this causes a great economic and social burden that alerts to the need for prevention and treatment of lean tissue loss, especially in the older diabetic population, which allows preserving functionality and maintaining a good quality of life for older adults who suffer from this sickness. The main therapy, both in the prevention and in the treatment of sarcopenia, is lifestyle modification and it is even expected that it will continue to be so in the future despite the possible approval of pharmacological agents due to their great scientific evidence [66]. Multicomponent physical exercise in combination with a nutritional intervention with special emphasis on protein intake are the bases of this intervention [67,68].

In addition, the new molecules used in the treatment of T2D, such as sodium–glucose cotransporter 2 (SGLT-2) inhibitors and glucagon-like peptide 1 receptor (GLP-1) agonists, also have an effect on body composition that should be considered at the time of treatment initiation, especially in older adults [69]. Although more scientific evidence is needed since the use of these drugs is relatively recent, a loss of the lean compartment has been seen in different studies. Several authors have confirmed an increased risk of sarcopenia and a loss of LM in individuals with T2D treated with SGLT-2 inhibitors [70,71]. Others, however, have shown maintenance of muscle mass during treatment [72,73]. The favorable effect of these drugs on body weight and fat mass is evident, however, it is important not only to consider this positive result but also the impact it may have on functional mass.

However, our study has some limitations. First, it is a small cohort that is probably not representative of the geriatric population with T2D, since they present good glycemic control and optimal nutritional status, reflected both in serum albumin and in the MNA nutritional screening. Secondly, due to the lack of a non-diabetic control group that would allow these results to be compared. The obtained results, despite these limitations, have important implications since they can open up new therapeutic possibilities in the face of one of the challenges of the current health system: the high prevalence of T2D in elderly.

## 5. Conclusions

The presence of low LM, both reflected in the form of FFM and in the form of SMM, in the geriatric population diagnosed with T2D is associated with poorer glycemic control and greater OS assessed by MDA. However, the assessment of body composition is not routinely performed in these patients, so we believe that this practice should be included for the detection of low LM and its prevention and/or treatment should be considered in diabetic older adults.

## Figures and Tables

**Figure 1 nutrients-15-03167-f001:**
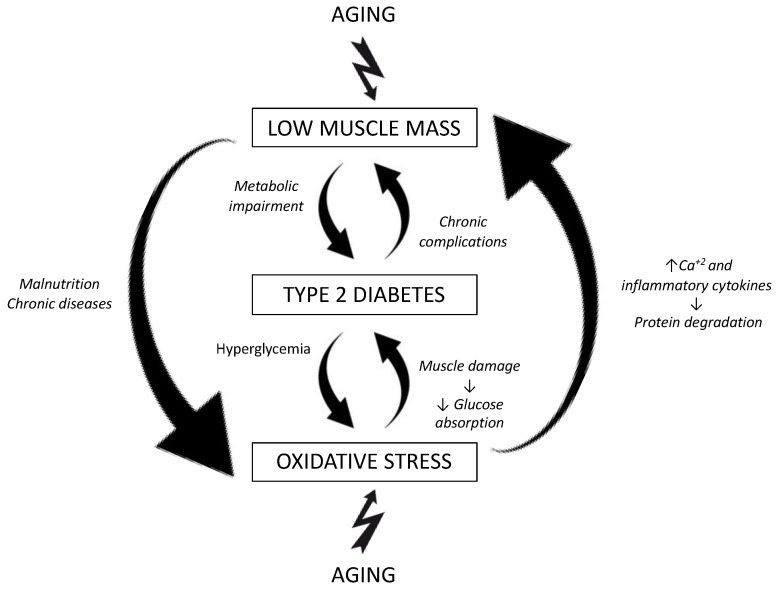
Relation between muscle mass, oxidative stress, aging, and type 2 diabetes mellitus.

**Figure 2 nutrients-15-03167-f002:**
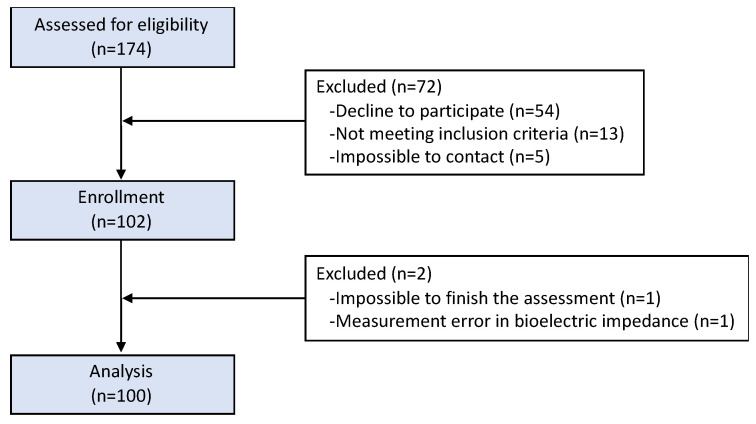
CONSORT diagram of participant through the study.

**Table 1 nutrients-15-03167-t001:** Characteristics of the patients with type 2 diabetes included in the study.

Characteristic	Total (*n* = 100)
Age (years)	70.3 ± 3.8
Time of T2D evolution (years)	17.8 ± 10.7
Body mass index (kg/m^2^)	30.8 ± 4.2
Brachial circumference (cm)	31.5 ± 3.4
Calf circumference (cm)	37.2 ± 3.1
Fat mass index (kg/m^2^)	8.9 ± 3.8
Fat-free mass index (kg/m^2^)	21.7 ± 2.6
Skeletal muscle mass index (kg/m^2^)	10.1 ± 1.9
Appendicular skeletal muscle mass index (kg/m^2^)	7.9 ± 1.0
Maximum muscle strength (kg)	28.6 ± 10.0
Gait speed (m/s)	0.8 ± 0.2
Protein intake (%)	20.0 ± 4.2
MNA score	26.5 ± 2.1
Glucose (mg/dL)	148.2 ± 45.7
HbA1c (%)	7.4 ± 1.1
Total cholesterol (mg/dL)	156.1 ± 30.2
LDL cholesterol (mg/dL)	93.0 ± 22.6
HDL cholesterol (mg/dL)	47.7 ± 12.8
Triglycerides (mg/dL)	134.5 ± 62.3
Albumin (g/dL)	4.2 ± 0.2
C-reactive protein	4.5 ± 8.9
Malondialdehyde (µM)	6.0 ± 5.8
Protein carbonyls (U.A.)	81.3 ± 20.9

Data are shown as average ± standard deviation. T2D = type 2 diabetes. MNA = mini nutritional assessment. HbA1c = glycated hemoglobin. LDL = low-density lipoprotein cholesterol. HDL = high-density lipoprotein cholesterol.

**Table 2 nutrients-15-03167-t002:** Characteristics of the patients with type 2 diabetes included in the study sorted by gender.

Characteristic	Women (*n* = 52)	Men (*n* = 48)
Age (years)	70.6 ± 3.6	70.0 ± 4.0
Time of T2D evolution (years)	20.2 ± 11.2	15.3 ± 9.7 *
Body mass index (kg/m^2^)	31.0 ± 4.4	30.5 ± 4.2
Brachial circumference (cm)	32.1 ± 3.7	30.9 ± 3.0
Calf circumference (cm)	36.7 ± 3.1	37.9 ± 3.1 *
Fat mass index (kg/m^2^)	10.7 ± 3.8	7.0 ± 2.7 *
Fat-free mass index (kg/m^2^)	20.1 ± 1.8	23.3 ± 2.3 *
Skeletal muscle mass index (kg/m^2^)	8.6 ± 1.0	11.6 ± 1.4 *
Appendicular skeletal muscle mass index (kg/m^2^)	7.3 ± 0.8	8.6 ± 0.9 *
Maximum muscle strength (kg)	21.0 ± 4.4	36.9 ± 7.6 *
Gait speed (m/s)	0.86 ± 0.2	0.71 ± 0.2 *
Protein intake (%)	20.3 ± 4.3	19.8 ± 4.0
MNA score	25.8 ± 2.2	27.3 ± 1.7 *
Glucose (mg/dL)	147.1 ± 47.7	149.3 ± 43.7
HbA1c (%)	7.3 ± 1.0	7.4 ± 1.1
Total cholesterol (mg/dL)	165.7 ± 29.9	145.4 ± 27.1 *
LDL cholesterol (mg/dL)	96.4 ± 23.6	89.1 ± 21.1
HDL cholesterol (mg/dL)	51.6 ± 14.4	43.4 ± 9.2 *
Triglycerides (mg/dL)	141.4 ± 57.4	126.8 ± 67.0
Albumin (g/dL)	4.2 ± 0.2	4.2 ± 0.3
C-reactive protein	4.2 ± 6.7	4.8 ± 10.9
Malondialdehyde (µM)	6.1 ± 5.6	5.8 ± 6.0
Protein carbonyls (U.A.)	84.5 ± 26.3	78.0 ± 12.9

Data are shown as average ± standard deviation. * *p* < 0.05 between women and men. T2D = type 2 diabetes. MNA = mini nutritional assessment. HbA1c = glycated hemoglobin. LDL = low-density lipoprotein cholesterol. HDL = high-density lipoprotein cholesterol.

**Table 3 nutrients-15-03167-t003:** Characteristics of the cohort according to fat-free mass index percentiles.

Characteristic	Total (*n* = 100)		Women (*n* = 52)		Men (*n* = 48)	
	≤25th Percentile (*n* = 25)	>25th Percentile (*n* = 74)	≤25th Percentile (*n* = 11)	>25th Percentile (*n* = 39)	≤25th Percentile (*n* = 12)	>25th Percentile (*n* = 36)
Time of T2D evolution (years)	19.7 ± 12.2	17.2 ± 10.4	24.2 ± 13.4	18.9 ± 10.4	14.9 ± 8.8	15.4 ± 10.2
Brachial circumference (cm)	29.3 ± 2.8 *	32.1 ± 3.3	30.1 ± 3.5 *	32.7 ± 3.7	28.2 ± 1.2 *	31.4 ± 2.7
Calf circumference (cm)	35.5 ± 2.2 *	37.7 ± 3.2	36.4 ± 2.1	36.7 ± 3.4	34.6 ± 1.9 *	38.7 ± 2.6
Maximum muscle strength (kg)	28.4 ± 11.2	28.8 ± 9.8	21.1 ± 5.4	21.0 ± 4.4	36.4 ± 10.6	37.0 ± 6.7
Gait speed (m/s)	0.80 ± 0.26	0.79 ± 0.18	0.83 ± 0.26	0.88 ± 0.19	0.78 ± 0.27	0.70 ± 0.09
Protein intake (%)	18.2 ± 4.1 *	20.7 ± 4.0	18.3 ± 4.8 *	21.0 ± 4.1	18.1 ± 3.5	20.4 ± 4.0
MNA score	26.2 ± 2.2	26.6 ± 2.1	25.7 ± 2.2	25.7 ± 2.2	26.8 ± 2.2	27.4 ± 1.5
Glucose (mg/dL)	160.6 ± 52.0	143.9 ± 43.1	178.3 ± 41.1 *	136.8 ± 45.2	141.3 ± 57.4	151.3 ± 40.0
HbA1c (%)	7.6 ± 1.4	7.3 ± 0.9	7.8 ± 1.1 *	7.2 ± 0.9	7.3 ± 1.6	7.4 ± 1.0
Malondialdehyde (µM)	7.1 ± 6.5	5.7 ± 5.6	8.0 ± 7.2	5.6 ± 5.1	6.2 ± 6.0	5.7 ± 6.1
Protein carbonyls (U.A.)	76.5 ± 15.1	84.3 ± 21.4	76.8 ± 18.6	88.7 ± 27.5	76.1 ± 11.6	79.9 ± 11.4

Data are shown as average ± standard deviation. * *p* < 0.05 vs. >25th percentile in the same group. T2D = type 2 diabetes. MNA = mini nutritional assessment. HbA1c = glycated hemoglobin.

**Table 4 nutrients-15-03167-t004:** Characteristics of the cohort according to skeletal muscle mass (kg) percentiles.

Characteristic	Total (*n* = 100)		Women (*n* = 52)		Men (*n* = 48)	
	≤25th Percentile (*n* = 25)	>25th Percentile (*n* = 74)	≤25th Percentile (*n* = 11)	>25th Percentile (*n* = 39)	≤25th Percentile (*n* = 12)	>25th Percentile (*n* = 36)
Time of T2D evolution (years)	17.0 ± 9.2	18.1 ± 11.3	18.4 ± 9.0	21.0 ± 11.9	15.5 ± 9.6	15.2 ± 9.9
Brachial circumference (cm)	30.7 ± 4.0	31.8 ± 3.2	32.0 ± 4.5	32.1 ± 3.5	29.0 ± 2.7 *	31.4 ± 2.9
Calf circumference (cm)	35.8 ± 2.3 *	37.7 ± 3.2	36.1 ± 2.2	36.8 ± 3.4	35.4 ± 2.6 *	38.6 ± 2.8
Maximum muscle strength (kg)	26.0 ± 10.5	29.6 ± 9.8	19.1 ± 3.6	21.6 ± 4.5	33.5 ± 10.4	38.0 ± 6.1
Gait speed (m/s)	0.79 ± 0.20	0.79 ± 0.20	0.80 ± 0.16	0.89 ± 0.22	0.78 ± 0.25	0.69 ± 0.09
Protein intake (%)	19.0 ± 4.6	20.4 ± 4.0	19.9 ± 5.5	20.4 ± 4.0	18.1 ± 3.4	20.3 ± 4.0
MNA score	26.6 ± 2.5	26.5 ± 1.9	26.1 ± 2.9	25.7 ± 1.9	27.1 ± 2.1	27.4 ± 1.6
Glucose (mg/dL)	153.0 ± 50.1	147.2 ± 44.3	167.6 ± 49.6	141.3 ± 45.9	137.3 ± 47.5	153.3 ± 42.4
HbA1c (%)	7.5 ± 1.2	7.3 ± 1.0	7.9 ± 1.1 *	7.2 ± 0.9	7.2 ± 1.3	7.5 ± 1.1
Malondialdehyde (µM)	9.0 ± 7.3 *	4.9 ± 4.8	9.0 ± 7.1 *	5.0 ± 4.7	9.1 ± 7.7 *	4.7 ± 4.9
Protein carbonyls (U.A.)	78.4 ± 17.4	82.8 ± 21.9	77.0 ± 21.9	88.3 ± 27.3	79.9 ± 11.8	77.5 ± 13.4

Data are shown as average ± standard deviation. * *p* < 0.05 vs. >25th percentile in the same group. T2D = type 2 diabetes. MNA = mini nutritional assessment. HbA1c = glycated hemoglobin.

**Table 5 nutrients-15-03167-t005:** Characteristics of the cohort according to skeletal muscle mass index percentiles.

Characteristic	Total (*n* = 100)		Women (*n* = 52)		Men (*n* = 48)	
	≤25th Percentile (*n* = 25)	>25th Percentile (*n* = 74)	≤25th Percentile (*n* = 11)	>25th Percentile (*n* = 39)	≤25th Percentile (*n* = 12)	>25th Percentile (*n* = 36)
Time of T2D evolution (years)	19.3 ± 11.7	17.2 ± 10.4	24.2 ± 13.1	18.8 ± 10.3	14.7 ± 8.4	15.5 ± 10.4
Brachial circumference (cm)	30.8 ± 4.5	31.8 ± 2.9	32.7 ± 5.3	31.9 ± 3.0	29.0 ± 2.5 *	31.7 ± 2.8
Calf circumference (cm)	35.7 ± 2.1 *	37.8 ± 3.3	36.1 ± 2.0	36.8 ± 3.4	35.3 ± 2.2 *	39.0 ± 2.7
Maximum muscle strength (kg)	27.8 ± 10.6	29.0 ± 9.9	20.0 ± 5.0	21.3 ± 4.2	35.1 ± 9.2	37.7 ± 6.7
Gait speed (m/s)	0.81 ± 0.23	0.78 ± 0.18	0.88 ± 0.22	0.86 ± 0.21	0.74 ± 0.23	0.70 ± 0.10
Protein intake (%)	19.3 ± 4.6	20.3 ± 4.0	20.8 ± 5.6	20.1 ± 3.9	18.0 ± 3.1 *	20.6 ± 4.1
MNA score	26.2 ± 2.3	26.6 ± 2.0	25.7 ± 2.4	25.8 ± 2.1	26.7 ± 2.0	27.5 ± 1.5
Glucose (mg/dL)	158.0 ± 51.2	144.8 ± 42.9	164.0 ± 46.1	142.0 ± 47.6	152.4 ± 56.6	147.9 ± 37.4
HbA1c (%)	7.5 ± 1.2	7.3 ± 1.0	7.8 ± 1.1 *	7.2 ± 0.9	7.3 ± 1.3	7.5 ± 1.1
Malondialdehyde (µM)	7.8 ± 6.2 *	5.2 ± 5.5	8.4 ± 6.6 *	5.1 ± 5.0	7.1 ± 6.0	5.2 ± 6.0
Protein carbonyls (U.A.)	79.7 ± 18.6	82.5 ± 21.9	80.9 ± 24.0	87.0 ± 27.2	78.6 ± 12.3	77.9 ± 13.4

Data are shown as average ± standard deviation. * *p* < 0.05 vs. >25th percentile in the same group. T2D = type 2 diabetes. MNA = mini nutritional assessment. HbA1c = glycated hemoglobin.

**Table 6 nutrients-15-03167-t006:** Multivariate logistic regression analysis for fat-free mass index (**A**) and muscle mass (**B**) parameters.

**A. Dependent Variable: Fat-Free Mass Index**	** *B* **	**SE**	**Wald**	**Significance**	**Exp(B)**
(Constant)	−1.431	4.779	0.090	0.765	0.239
Brachial circumference (cm)	0.191	0.118	2.638	0.104	1.211
Glucose (mg/dL)	−0.017	0.010	3.122	0.077	0.983
HbA1c (%)	−0.097	0.428	0.051	0.821	0.908
**B. Dependent Variable: Skeletal Muscle Mass**	** *B* **	**SE**	**Wald**	**Significance**	**Exp(B)**
(Constant)	7.583	3.235	5.494	0.019	1964.920
Malondialdehyde (µM)	−0.142	0.064	4.881	0.027	0.868
Glucose (mg/dL)	−0.002	0.009	0.049	0.829	0.998
HbA1c (%)	−0.703	0.466	2.278	0.131	0.495

## Data Availability

All data are available from the corresponding author upon reasonable request.

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
