# Peer review of "Low Muscle Mass Is Associated with Poorer Glycemic Control and Higher Oxidative Stress in Older Patients with Type 2 Diabetes"

_nutrients, 2023, doi:10.3390/nu15143167_

Round 1
Reviewer 1 Report
Authors made research regarding Low muscle mass is associated with poorer glycemic control and higher oxidative stress in older patients with type diabetes. Please see my suggestions bellow:
Introduction is too poor.
L56-58. L97-100. Aim of the study is 2.5 lines. It needs to be addressed from the perspective of describing the contribution to the field under analysis and the elements of scientific novelty presented. Develop it better. Make the reader to understand the importance of your research.
Before introducing any Figure /Table, it must be mentioned in the main text. Apply for Figure 1.
2.1. CONSORT flow chart should be presented regarding criteria for patients’ selection, instead of sending the reader to refs. 9, 10.
L78. No/date must be provided for the Ethical approval. “Ethics Committee of our center” – the center must be detailed.
L81. Which “other things”? this is not a scientific statement.
L94. Authors must provide the Model, Producer/manufacturer, City and Country for EACH APPARATUS used in the research. For “A single-frequency bioe-94 lectric impedance” – the model is missing. Check all the apparatus mentioned in your research and complete their info (BD 106 Vacutainer tube, HPLC, centrifuge, etc.).
L106. The Producer, Country, purity degree, and concentration used for each REAGENT/chemical used must be also provided. Check all the chemicals/reagents mentioned in your research and complete their info.
According to the Instructions for authors, Acronyms/Abbreviations/Initialisms should be defined the first time they appear in each of three sections: the abstract; the main text; the first figure or table. When defined for the first time, the acronym/abbreviation/initialism should be added in parentheses after the written-out form.
2.6. All computer programs/softs used, and their variants must be mentioned and referenced. Please check and complete.
L138. Duplicate info with L166.
Complete the head of the tables 1 to 4, for the first column (Characteristics). They are not allowed empty cells in a scientific table. Abbreviations used in those tables must be explained under them.
The Discussion chapter needs to be improved. Please describe potential diet therapies that can increase muscle mass in older patients and their impact on oxidative stress https://doi.org/10.1016/j.biopha.2022.113238 ; I consider referring to the ketogenic diet and to low-carb diet – check PMID: 33121986.
Regarding the statistics, you must make univariate and multivariate regression and see whether the low muscle mass is still associated with poor glycaemic control.
The role of novel molecules (such as SGLT-2 inhibitors and GLP-1 agonists) on the muscle mass should be also considered – I suggest checking and referring to https://doi.org/10.3390/ijms24044029 and PMID: 32765722.
Some references are very old. Try to up-date that section.
Good English.
Reviewer 2 Report
In this article, the authors studied differences in clinical and nutritional parameters, disease control, and oxidative stress in a cohort of elderly patients with type 2 diabetes. Since the paper could be of interest to the scientific community, I'd recommend proceed after some minor revisions of the manuscript:
1. None of the manuscript's authors has the affiliation marked with the number 7. Please check and correct it!
2. In line 27, before the abbreviation "LM" write its full name!
3. In line 113, you wrote that the blood sample was centrifuged at 1500 rpm. Instead of the rpm unit, I advise the authors to express the centrifugation in the rcf (g) unit!
4. In line 114: correct „ 4°C“ to „4 °C“, and “-20ºC” to “-20 ºC”
5. In line 125: correct „5%“ to „5 %“, and “20%” to “20 %”. Similar typographical errors can be found in other places in the text of the manuscript, so I suggest the authors review the entire paper and correct the errors carefully.
Round 2
Reviewer 1 Report
The authors have significantly improved the manuscript based on the suggestions received.